# Optimization and Application of the QuEChERS-UHPLC-QTOF-MS Method for the Determination of Broflanilide Residues in Agricultural Soils

**DOI:** 10.3390/molecules29071428

**Published:** 2024-03-22

**Authors:** Xiaoli Nie, Guai Xie, Zhitao Huo, Baoyu Zhang, Haifei Lu, Yi Huang, Xin Li, Liangliang Dai, Siyuan Huang, Ailin Yu

**Affiliations:** 1Changsha General Survey of Natural Resources Centre, China Geological Survey, No. 258 Xuefu Road, Suburban Street, Changsha 410000, China; niexiaoli@mail.cgs.gov.cn (X.N.); niexiaoli0912@163.com (Z.H.); lixin@mail.cgs.gov.cn (X.L.); dailiangliang@mail.cgs.gov.cn (L.D.); 2Jiangxi Academy of Forestry, No. 1629 West Fenglin Road, Economic and Technological Development Area, Nanchang 330000, China; 18379995628@163.com (G.X.); 13260679363@163.com (B.Z.); huangsiyuan9959@163.com (S.H.); 3College of Urban Construction, Zhejiang Shuren University, No. 8 Shuren Road, Gongshu District, Hangzhou 310015, China; 4School of Pharmacy, Jiangxi University of Traditional Chinese Medicine, No. 1688 Meiling Road, Xinjian District, Nanchang 330000, China

**Keywords:** broflanilide, residue analysis, soil, optimization

## Abstract

In this study, the separation conditions of UHPLC-QTOF-MS and the extraction conditions of QuEChERS were optimized. The analytical process for determining Broflanilide residues in different soil types was successfully established and applied to its adsorption, desorption, and leaching in soil. Broflanilide was extracted from soil with acetonitrile and purified using PSA and MgSO_4_. The modified UHPLC-QTOF-MS method was used for quantification. The average recovery of Broflanilide was between 87.7% and 94.38%, with the RSD lower than 7.6%. In the analysis of adsorption, desorption, and leaching quantities in four soil types, the RSD was less than 9.2%, showing good stability of the method, which can be applied to determine the residue of Broflanilide in different soils.

## 1. Introduction

The global population is expected to increase to nine billion by 2050 [1]. In the foreseeable future, agricultural production must be greatly increased to accommodate the population growth with increased animal farming for food and feed. Hence, food security will become an urgent key issue. Using chemicals can protect plants against insects, animals, and weeds in farming [2,3]. However, pesticides are environmentally exogenous and enter the environment by spraying, seed sowing, fumigation, or direct dressing. The accumulation of residual persistent pesticides can adversely affect human health and ecosystems [4]. Soil can act as a ‘reservoir’ and ‘distribution center’ of chemical pesticides, which may hold about 50% of the residual persistent pesticides [5]. Although they can be gradually dissipated through crop absorption, evaporation, and degradation, the decreasing rates are normally slower than the agricultural cycle [6]. In addition, pesticides in soil can also form bound residues, which can be absorbed by crop roots or gradually leached into groundwater. Either way, pesticide contamination threatens the soil’s ecological balance and human long-term health [7,8].

Broflanilide [3-benzamido-N-(4-(perfluoropropan-2-yl) phenyl) benzamide], known as Broflanilide (Table 1), is a new class of organic halide pesticide containing diamide. It was jointly invented by Mitsui Chemical Agricultural Co., Ltd. (Tokyo, Japan) and BASF (Ludwigshafen, Germany).

Broflanilide shows potent insecticidal properties against various pests, including beetles, termites, caterpillars, and other common insects found on cereals, leafy vegetables, and perennial plants [9]. Research indicates that it produces desmethyl-broflanilide through metabolism. Broflanilide acts as a noncompetitive antagonist against the resistant-to-dieldrin γ-aminobutyric acid receptor. Interestingly, desmethyl-broflanilide has a binding mechanism different from that of traditional noncompetitive antagonists, such as Fipronil [10]. Broflanilide formulations have received approval for use in several countries, including Japan (2018), Canada (2020), Australia (2020), China (2020), and the United States (2021) [11]. Consequently, broflanilide is poised to play a significant role in pest management, particularly against pests resistant to conventional noncompetitive antagonists [10].

Pesticide multi-residue analysis methods include sample preparation, pesticide extraction, and separation, followed by quantitative analysis. The QuEChERS method was first proposed at the European Pesticide Residues Symposium (EPRW) in Rome in 2002. Anastassiades et al. published this method in 2003 [12], which has since become widely adopted as a standard sample preparation technique in the analysis of pesticide residues. The QuEChERS method is suitable for a variety of pesticides. It reduces the use of organic solvents and soil samples since it has a higher recovery rate [13]. UHPLC-QTOF-MS is also recognized as an emerging technology for analyzing pesticide residues in food, mainly due to its accuracy. It offers a balanced combination of high-resolution precision with sensitive full-scan capabilities. Thus, QTOF complements other mass spectrometers like quadrupole and ion trap instruments for identification and quantification [14,15,16]. In this study, it was utilized ultra-high performance liquid chromatography-time of flight-mass spectrometry (UHPLC-QTOF-MS) in conjunction with the modified QuEChERS method to enhance the response and recovery. It optimized the conditions and parameters of UHPLC-QTOF-MS by comparing various solvents and adsorbents. Consequently, the main objective of this research was to establish an efficient and reliable method for detecting broflanilide residues in soil samples.

## 2. Results and Discussion

### 2.1. Optimization of Separation Conditions of UHPLC-QTOF-MS

#### 2.1.1. Chromatographic Column Optimization

The choice of the chromatographic column is critical in inferencing analyte retention behavior and peak shape [17]. Specifically, selecting an appropriate chromatographic column can enhance the stability and efficiency of the analysis. The most commonly used chromatographic column in determining pesticide compounds is octadecyl-bonded C18 [18]. To reduce the running time and achieve optimal peak shapes, we employed two distinct C18 columns for optimizing the UHPLC conditions: the Zorbax Eclipse XDB-C18 column (4.6 mm × 150 mm × 5 μm) and the Zorbax SB-C18 column (2.1 mm × 100 mm × 1.8 μm). With identical mobile phases, the retention time of broflanilide on the XDB-C18 column is longer, suggesting a superior separation between broflanilide and impurity peaks. Consequently, the XDB-C18 column was selected for further study.

#### 2.1.2. Mobile Phase Optimization

To enhance the performance of UHPLC, the impact of flow rate and mobile phase on broflanilide separation was investigated. The compared mobile phases are methanol-water and acetonitrile-water (with or without 0.1% formic acid). The acetonitrile-water phases produced faster separation with narrower peaks. This is likely because acetonitrile exhibits stronger solvation tendencies towards solutes than methanol [19,20,21], and acetonitrile–water mixtures demonstrate higher solvent permeability [22], effectively removing impurities in the column.

Adding formic acid (0.1%) optimized peak shapes, enhancing sensitivity [23]. However, adding formic acid (0.1%) in acetonitrile–water led to prolonged retention times due to the low acetonitrile ratio, delaying the detection of subsequent samples. Conversely, a high acetonitrile proportion resulted in shorter retention times, introducing impurity peak interference. When using acetonitrile–water with formic acid (0.1%) at a 75:25 volume ratio at a flow rate of 0.5 mL/min, clear separation was achieved between the broflanilide and the impurity peaks. A symmetric peak was obtained with a stable baseline and a shortened retention time of 2.08 min. Hence, this mobile phase offers the optimal composition for the analysis of broflanilide in HPLC. Figure 1 depicts the UHPLC chromatogram of the standard broflanilide sample.

### 2.2. QuEChERS Method Optimization

#### 2.2.1. Solvent Optimization

The most time-consuming and intricate process in residual pesticide analysis involves the pretreatment of raw samples. The chemical property of the insecticides dictates the choice of solvent for extraction. Commonly employed solvents include acetone [24], acetonitrile [25], methanol [26], ethyl acetate, and dichloromethane [27]. Here, the recovery rates for the paddy soil samples were compared using these solvents. Figure 2 shows that the highest recovery rates were obtained from acetonitrile, reaching 96.73% at a spiked concentration of 0.01 mg/L and 95.34% at a spiked concentration of 0.1 mg/L. Meanwhile, compared with other solvents, acetonitrile offers several additional advantages in pesticide residue analysis, such as minimal interference from lipids and proteins, high compatibility with UHPLC systems, and fewer co-extracted matrix components [28].

In order to determine whether formic acid and sodium hydroxide have an optimization effect on the recovery, they were added into acetonitrile at 0.1% vol. As shown in Figure 2, there is no improved extraction of Broflanilide in paddy soil, indicating that the acidic or alkaline conditions have no benefit in Broflanilide extraction. The ionization state of Broflanilide in the presence of formic acid and ammonia in acetonitrile and its solubility in the extraction medium may be the reason for this result [29]. Consequently, unmodified acetonitrile was used.

#### 2.2.2. Adsorbent Optimization

Satisfactory extraction of broflanilide diamide was observed for the C18, GCB, and PSA adsorbents using the QuEChERS method. As an anion exchanger, PSA extracts carbohydrates and acids from the mixed sample [30]. C18, on the other hand, is used to extract non-polar and medium-polar compounds due to its non-polar properties. GCB is an effective adsorbent for pigments, such as carotenoids and chlorophyll [31]. Adding MgSO_4_ and NaCl improves the partition and generates a better phase separation with higher recovery rates [32]. This study used 50 mg of each specific adsorbent (0.1 mg/kg) for broflanilide extraction with added 150 mg MgSO_4_. As indicated in Table 2, the average recovery rates of various adsorbents in rice soil ranged from 89.85% to 93.81%, satisfying the criteria of the Chinese national agricultural standard between 70% and 120% [33]. Notably, the highest recovery rate of 93.81% was achieved from PSA.

The appropriate broflanilide extraction and purification conditions via the QuEChERS procedure are identified with 50 mg of PSA combined with 150 mg of MgSO_4_.

### 2.3. Verification of the Method

#### 2.3.1. Linearity, Specificity, LOD, LOQ, and Matrix Effects

No interference was observed in the blank with various matrices throughout the retention time, indicating that the proposed method had high specificity for Broflanilide. The 5-point calibration plots (31.25, 100, 250, 500, and 1000 μg/kg) were obtained by least squares linear regression, as shown in Figure 3. A good linear relationship was obtained in the matrix test (correlation coefficient >0.999). The Quantification limit (LOQ) and detection limit (LOD) of Broflanilide in soil samples were 5.94 and 1.25 μg/kg, respectively. The matrix effect value in the blank soil sample was −58%. The results showed that broflanilide had a signal inhibition effect in soil samples [34].

#### 2.3.2. Accuracy and Precision

Broflanilide was added into blank soil samples at the concentrations of 0.1, 0.5, and 1.0 mg/kg, and the measured recovery rate and the relative standard deviation (RSD) are presented in Table 3. The average broflanilide recovery ranged from 87.7% to 92.91%, with RSD values between 5.49% and 7.51%. These results demonstrate the accuracy and precision of the established method.

### 2.4. Application in Practical Research

#### 2.4.1. Applicability Test

Broflanilide was added to different soil samples at the contents of 0.0, 0.1, 0.5, and 1.0 mg/kg before the adsorption–desorption and leaching experiments and used the improved method for testing. The results in Figure 4 demonstrate the average broflanilide recovery rate was 87.70–94.38% at different spiked concentrations. The RSD distribution map (Figure 4b) showed that the RSD value was 1.70–7.53%. This shows that our method offers high-quality accuracy and precision in different types of soils and can be used to analyze the concentration of broflanilide in different types of soils. 

#### 2.4.2. Application of Soil Adsorption-Desorption

The results of measuring the soil in the adsorption–desorption process are displayed in Figure 5. Phaeozems can better adsorb broflanilide, and it is less likely to be desorbed, determined by its large organic content. From the literature, the adsorption of pesticides in soil is affected by the concentrations of organic composition in soil [35,36,37]. According to the RSD distribution map of the test results, it can be seen that the RSD is lower than 9.2%, so the method discussed in this paper has good stability and can be well applied to soil adsorption–desorption research and detection of broflanilide.

#### 2.4.3. Application in Soil Leaching Experiment

Leaching refers to the vertical downward movement of pesticides when aquatic solutions penetrate the soil. This process involves the comprehensive behavior of pesticide distribution, desorption, and adsorption in conjunction with soil and water interactions [38]. Utilizing the soil column leaching method, we simulated the migration process of broflanilide in natural soil and obtained leaching data, as depicted in Figure 6. The concentration of broflanilide in different soil layers after leaching is related to its adsorption performance, which also determines the mobility of pesticides in soil [39]. Corresponding to the results of adsorption–desorption experiments, Phaeozems had better stability and could stabilize the broflanilide in the surface layer, while other soils did not have such ability. Most of the broflanilide in soil entered the leachate during the leaching process. The RSD of the leaching test results was 1.32–7.16%, indicating that our method offers good applicability and stability for detecting the leaching of broflanilide in soils.

## 3. Materials and Methods

### 3.1. Chemicals

Broflanilide (≥99.6%) and its solution (5%) were supplied by Mitsui Chemicals Agro Inc. (Tokyo, Japan). Graphitized carbon black (GCB) (60 μm) was purchased from Nanjing XFNANO Materials Tech Co., Ltd. (Nanjing, China). HCOOH (≥88%, chromatographic grade) was supplied by Tianjin Komio Chemical Reagent Co., Ltd. (Tianjin, China). CH_3_OH, CH_3_CN, ethyl acetate, (CH_3_)_2_CO, and CH_2_Cl_2_ with chromatographic grade were provided by Shantou Xilong Tech. Co., Ltd. (Shantou, China), who also supplied NaOH, NaN_3_, CaCl_2_, NaCl and anhydrous MgSO_4_. C18 (40 μm) and N-propyl ethylenediamine (PSA) (40 μm) were purchased from Tianjin Bonna-Agela Technologies (Tianjin, China). Deionized water was produced using a Milli-Q water purification system.

Broflanilide (101.4 mg) was ultrasonically dissolved in 85 mL acetonitrile in a volumetric flask (100 mL) for 20 min before standing for 40 min at room temperature (RT). The solution (1 mg/mL) was well shaken before being stored at 3 °C. To prepare the working solution, the broflanilide solution was diluted in acetonitrile to 31.25, 100, 250, 500, and 1000 μg/kg concentrations. The matrix-matching standard solution was prepared by adding the blank soil extract to the working solutions. The prepared solutions were refrigerated at −20 °C without light.

### 3.2. Instrument and Conditions

The broflanilide concentration was determined using ultra-high pressure liquid chromatography (UHPLC) coupled with a quadrupole time-of-flight mass spectrometer AB SCIEX X500R. An isocratic elution process used 40% solvent for 6 min at 0.3 mL/min. A 10 μL sample was injected and kept at 277.15 K.

The spray voltage of the electrospray ion source was 5500 V at 473.15 K. The ion transport tube was maintained at 823.15 K. The pressure of the ion source was 10 bar, while the pressures of the ion source 2 and curtain gas were set at 2 bar. The instrument operated in high-resolution multiple reaction monitoring (HR MRM) scanning mode, measuring the cations and covering a mass range of 100 to 1400 Da. The resolutions of the first and second mass were greater than 26,000 and 25,000 full widths at half maximum, respectively. The collision energy was maintained at 35 ± 15 eV.

### 3.3. Sample Collection and Pretreatment

The blank sample was obtained from a paddy field plot measuring 10 m × 6 m in Zengjia Village (E 115.1, N 28.3), Jiangxi, China. Samples were collected during the rice harvesting period. 1 kg of paddy soil was collected from each of the five random points at 0–20 cm depth and thoroughly mixed before being air-dried at RT. The soil samples were sieved using a sieve (20-mesh), and a 500 g sub-sample was prepared through four-fold extraction [40].

Following the ISO 10400-206 standard, fresh soils were collected from the farmland surface layer (0–15 cm) without pesticides, fertilizers, and biological additives in April 2023 from 4 different sampling locations detailed in Table 4. Soil samples were collected and transported in sealed vessels to maintain their initial properties. In the lab, all samples were air-dried at RT, lightly crushed, and passed through a 2 mm mesh sieve to remove large stones and debris. The left-over soils were stored at low temperatures (<276.15 K). The used samples were collected and processed by Huagen Environmental Group Co., Ltd. (Nanchang, China) to prevent pollution.

The physical and chemical characteristics of the soils were determined according to the World Soil Resource Reference (WRB) [41] and the standard protocol [42], as outlined in Table 4. Based on the soil classification system developed by the United Nations Food and Agriculture Organization [43], the four soils (S1 to S4) were classified as Luvisols, Anthrosols, Gleysols, and Phaeozems.

### 3.4. Validation of Method and Data Analysis

The QuEChERS method proposed here was validated following the guidelines specified by the International Pesticide Analysis Cooperation Committee. Various qualities, including linearity, specificity, accuracy, LOD, LOQ, precision, and matrix effect were comprehensively evaluated across various matrices [44,45].

The blank samples were analyzed to assess specificity. The lack of interference confirmed its specificity to broflanilide. The least squares regression was used to determine the R^2^, gradient, and intercept based on the measurement from various broflanilide concentrations of 31.25, 100, 250, 500, and 1000 μg/kg. Acceptance criteria for R^2^ were set at ≥0.99, and the goodness of fit was set at ≤20% [46]. The LOQ and LOD values were also calculated based on SN ratios of 10:1 and 3:1 for the spiked samples, respectively [47].

The matrix effect was assessed using pure CH_3_CN as the blank. Broflanilide standard solutions were prepared with various concentrations to produce a standard calibration curve in different matrices. The matrix effect constant (**ME**) was determined below [48]:$$\mathrm{ME}=(S_{m}/S_{s})\;\times 100\%$$
where *s* and *m* are the slopes of the calibration curve in the solvent and the matrix, respectively. **ME** > 10% represents the matrix-enhancing effect, while **ME** < −10% suggests the matrix-inhibiting effect. If the **ME** values are ±10%, the matrix effect is negligible. 

Accuracy and precision were assessed through recovery experiments. To determine accuracy, broflanilide solutions (0.1, 0.5, and 1.0 mg/kg (*n* = 5)) were added to blank soil samples on three occasions. The broflanilide recovery rate (%) was determined against the calibration curve. Acceptable recovery rates fell from 70% to 120% [49]. Precision was determined by the RSD of the broflanilide recovery rate. A value of less than 20% indicated high precision, otherwise it cannot be accepted. [50].

### 3.5. Method Application

#### 3.5.1. Applicability Experiment

According to the above recovery test method, the recovery experiments with spiked concentrations of 0.1, 0.5, and 1 mg/kg were carried out on four different soils on three dates, and the recovery rate and RSD of broflanilide were calculated and obtained.

#### 3.5.2. Adsorption-Desorption Experiment

The broflanilide adsorption–desorption in soil was investigated using the equilibrium oscillating method suggested by several reputable organizations, including the U.S. Environmental Protection Agency [51] and the Organization for Economic Co-operation and Development (OECD) [52], as well as a treatment method reported by Jiangxi Agricultural University [40,53]. To prepare the standard solution, 100 mg of NaN_3_ was dissolved in a 0.01 mol/L CaCl_2_ solution. To enhance phase separation and inhibit microbial degradation of broflanilide in soils, CaCl_2_ solution, and NaN_3_ solution, which mimic the ionic strength of natural soil solution, were used, respectively. The initial concentration of broflanilide was set to 1 mg/L. A 1:5 soil–water (*w*/*v* ratio) was maintained. In a centrifuge tube (50 mL), 2.0 g soil was added to 10 mL of the NaN_3_ and CaCl_2_ solution. The tubes were then oscillated at 180 rpm using a thermostatic oscillator at RT for varying durations of 0.5, 1, 2, 4, 6, 8, 12, 16, and 24 h to achieve equilibrium. The supernatant was separated, and the soil portion was collected. The soil was then extracted using the method described above and analyzed using UHPLC-QTOF-MS, and the ench sample was repeated 3 times.

After the adsorption experiment, the supernatant was discarded. In addition, CaCl_2_ injected the same amount of solution into the tube without pesticides. Afterward, the oscillator was used to oscillate the test tube at an ambient temperature of (298.15 ± 1 K). The subsequent operation and analysis were the same as the above adsorption experiment operation.

#### 3.5.3. Leaching Experiment

Under saturated flow conditions, the leaching behavior of broflanilide was studied in a filled soil column made of polyethylene plastic pipe (inner diameter 4 cm, height 30 cm), as shown in Figure 7. Two samples fortified with 250 and 500 μg were chosen following the OECD guidelines [52]. Before starting the leaching test, quartz sand (1 cm) and a layer of sand (G1 hole) were added to the bottom and top of the soil column to avoid soil disturbance. To eliminate any trapped air, the soil column was tightly packed with 500 to 700 g of soil. Subsequently, the reverse osmosis method utilizing a 0.01 mol/L CaCl_2_ solution was employed [54]. The remaining water was eliminated using a blow dryer. Then, 700 μL of a 100 mg/L broflanilide solution was infiltrated onto the soil surface. The oil sample was soaked for 24 h to reach adsorption equilibrium. For simulating rainfall leaching, 2000 mL of 0.01 mol/L CaCl_2_ solution was fed through the soil samples with a peristaltic pump operating for 96 h. The leachate was collected every 8 h in conical flasks. After completion of the leaching experiment, the soil column was removed and cut into three sections that were 10 cm long. The broflanilide was then extracted using the QuEChERS method, and the residue was quantified using UHPLC-QTOF-MS to determine the broflanilide content in both the leachate and soil components.

## 4. Conclusions

In this study, the separation conditions in the UHPLC-ESI-QTOF-MS method and the solvents and adsorbents in the QuEChERS method were optimized, and the specificity, linearity, accuracy, and precision of the method were successfully verified. Based on these works, an efficient, rapid, and sensitive method for the detection and analysis of the content of broflanilide in soil was established, and it obtained a good recovery rate when applied to the determination of the content of broflanilide in four types of soil at different spiked concentrations. This method showed excellent stability in the adsorption–desorption and leaching experiments applied to the soil. And the precise measurement results make it easy to analyze the behavior of broflanilide in soil. This study provides an effective method for the determination of the content of broflanilide in different types of soil and the experimental study of adsorption–desorption and leaching.

## Figures and Tables

**Figure 1 molecules-29-01428-f001:**
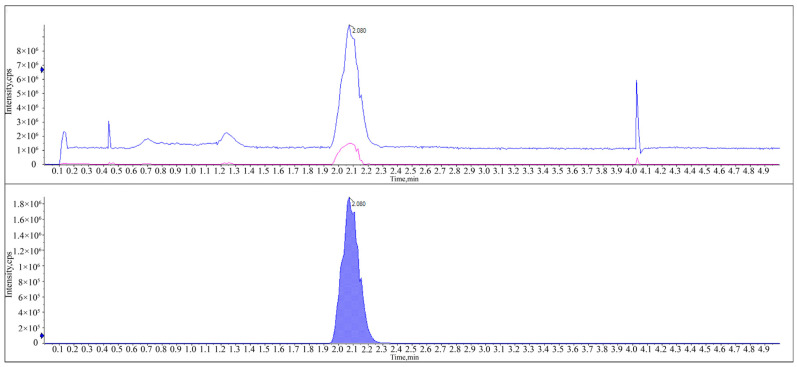
HPLC chromatograms of broflanilide detected in the standard solution (5 mg·kg^−1^).

**Figure 2 molecules-29-01428-f002:**
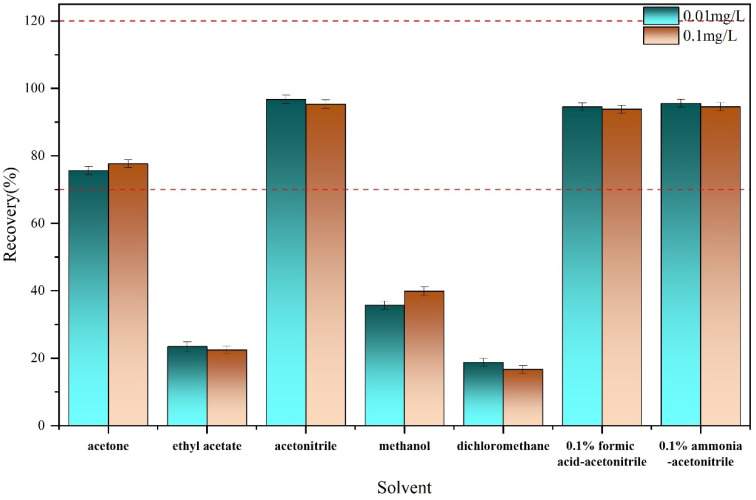
Effects of a variety of solvents (acetone, ethyl acetate, acetonitrile, methanol, dichloromethane, 0.1%formic acid-acetonitrile, and 0.1% ammonia- acetonitrile) on the recovery of broflanilide in paddy soil samples spiked at levels of 0.01 and 0.1 mg/L (*n* = 3).

**Figure 3 molecules-29-01428-f003:**
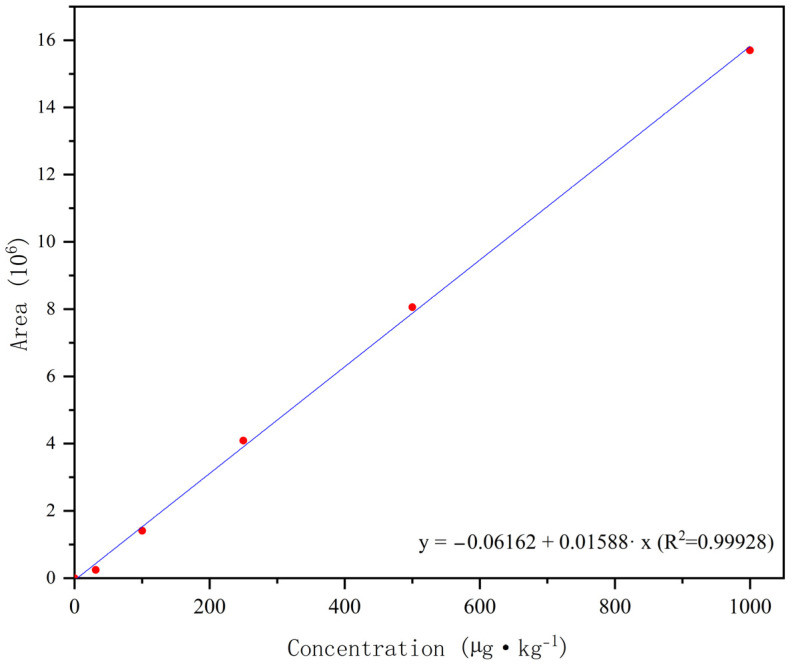
Linear calibration curve of broflanilide and its metabolites in paddy soil.

**Figure 4 molecules-29-01428-f004:**
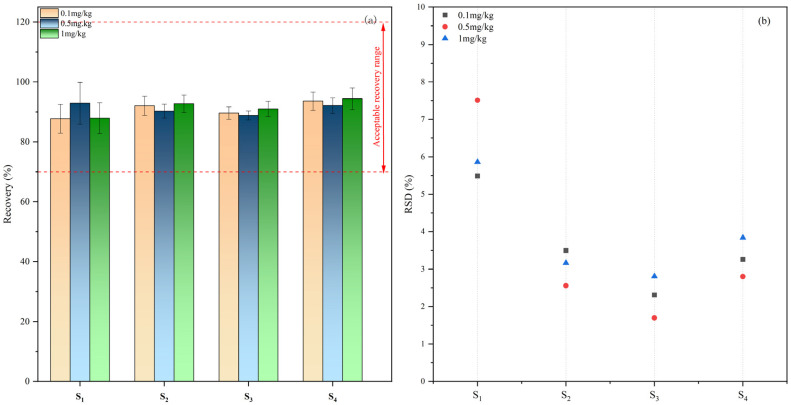
The recovery (**a**) and RSD distribution (**b**) of broflanilide in four agricultural soils.

**Figure 5 molecules-29-01428-f005:**
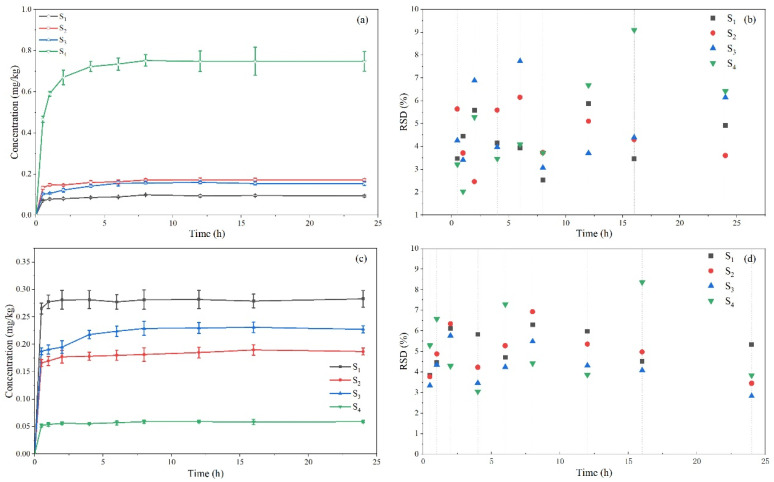
The content change curves of broflanilide in four different agricultural soils during adsorption (**a**) and desorption (**c**) and the RSD distribution graph during adsorption (**b**) and desorption (**d**).

**Figure 6 molecules-29-01428-f006:**
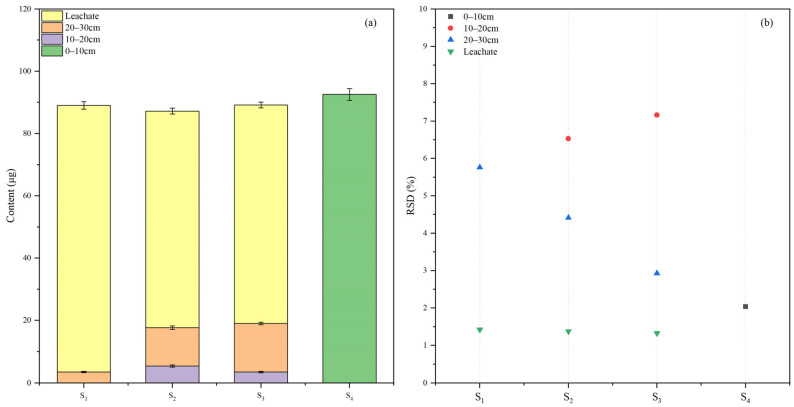
(**a**) distribution of broflanilide in soil column and leachate of four different agricultural soils; (**b**) The RSD value distribution graph of the test results was obtained.

**Figure 7 molecules-29-01428-f007:**
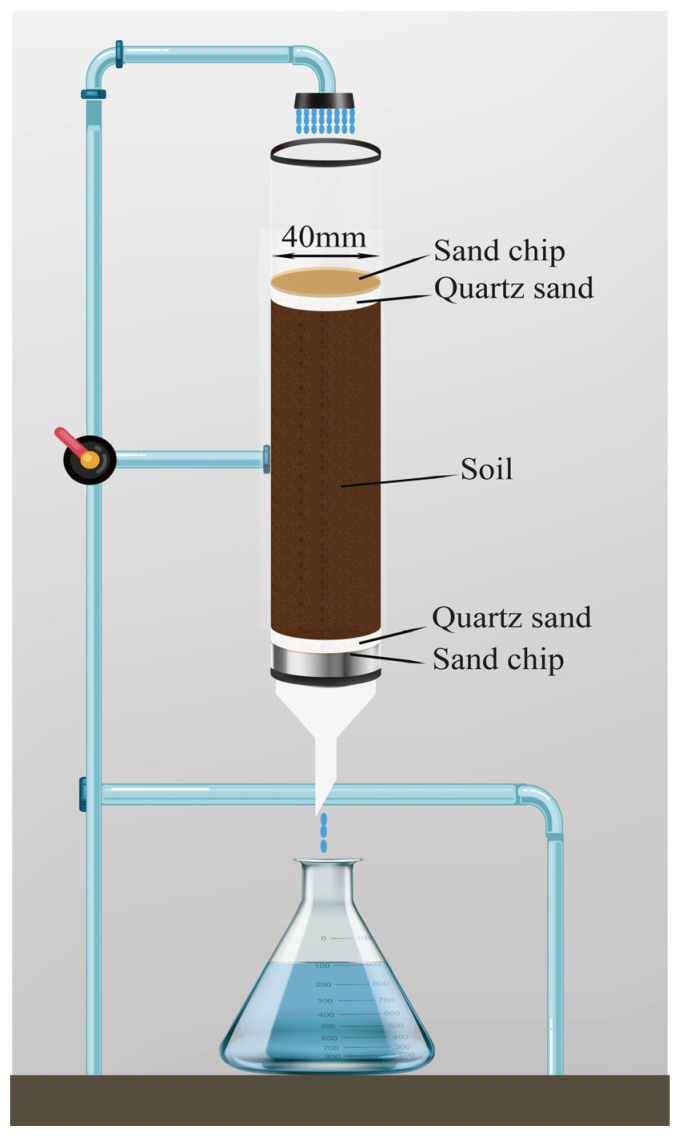
Sketch of soil column used in leaching experiment.

**Table 1 molecules-29-01428-t001:** Some important physicochemical properties of Broflanilide.

Chemical Structure	Item	Value
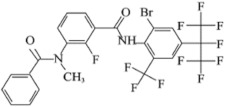	IUPAC ^a^ name	Broflanilide (BFL)
	Code name	MCI-8007
	Molecular formula	C_25_H_14_F_11_BrN_2_O_2_
	Molecular weight	663.28
	Formulation	WP 5%
	Density	1.7 g/cm^3^ (296.15 K)
	Solubility	0.71 mg/L (293.15 K, pure water)
	Vapor pressure	<9 × 10^−9^ Pa (298.15 K)
	Partition coefficient *Log P_ow_*	5.2 (293.15 K, pH 4)

^a^ IUPAC International Union of Pure and Applied Chemistry.

**Table 2 molecules-29-01428-t002:** Effects of different sorbents (PSA (primary secondary amine), N-propyl ethylenediamine; C18, octadecylsilane; and GCB, graphitized carbon black) on the recovery of broflanilide in paddy soil spiked at the level of 0.1 mg/L (or mg/kg; *n* = 3).

Sorbent	Recovery (%)			Average Value (%)
PSA	93.62	95.25	92.55	93.81
C18	92.73	93.56	91.55	92.61
GCB	89.69	88.32	91.54	89.85

**Table 3 molecules-29-01428-t003:** Recoveries and relative standard deviations of broflanilide spiked at three concentration levels each (*n* = 5).

Matrix	Spiked Level (mg/kg)	Recovery (%)					Average Recovery (%)	Relative Standard Deviation (%)
		1	2	3	4	5		
Paddy soil	0.1	89.36	80.02	92.54	86.46	90.14	87.70	5.49
	0.5	82.79	94.41	99.37	98.80	89.20	92.91	7.51
	1	89.73	80.54	94.64	86.15	88.45	87.90	5.86

**Table 4 molecules-29-01428-t004:** Sampling locations and physicochemical properties of four different agricultural soils.

Soil	Location (Latitude, Longitude)	Classification	Texture(%)				pH	CEC ^a^ (cmol/kg)	OC ^b^ (%)	OM ^c^ (%)
			Sand	Slit	Clay	Type				
S_1_	Shijiazhuang, Hebei (N 39°45′, E 117°32′)	Luvisols	40.62	35.91	23.47	Silt loam	6.58	26.15	0.14	0.48
S_2_	Ningbo, Zhejiang (N 29°14′, E 121°48′)	Anthrosols	47.98	23.18	28.84	Loam	7.85	12.90	0.92	1.66
S_3_	Chengdu, Sichuan (N 30°56′, E 105°51′)	Gleysols	45.83	40.83	13.34	Silt loam	8.48	25.40	0.07	0.17
S_4_	Haerbin, Heilongjiang (N 41°36′, E 127°53′)	Phaeozems	13.08	32.50	54.42	Sandy loam	6.38	30.36	2.08	4.64

^a^ CEC: Cation Exchange Capacity; ^b^ OC: Organic Carbon; ^c^ OM: Organic Matter.

## Data Availability

Data are contained within the article.

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
