# Peer review of "Optimization and Application of the QuEChERS-UHPLC-QTOF-MS Method for the Determination of Broflanilide Residues in Agricultural Soils"

_molecules, 2024, doi:10.3390/molecules29071428_

Round 1

Reviewer 1 Report

Comments and Suggestions for Authors

THE WORK MAY BE PUBLISHED. NEED TO SEE ATTENTION ALL CORRECTIONS AND SUGGESTIONS MARKED IN THE ATTACHED COPY OF THIS MANUSCRIPT.

Author Response

Thank you very much for your comments on this paper, so that this paper can be more refined. We carefully discussed your comments and made the following adjustments:
1.We have corrected the language problem in line 71,74,173 you proposed, please confirm.
2.In response to your questions about line75-77, we have highlighted the main purpose of the study.
3.For your comments in Line 128-129, we have demonstrated the relevant arguments and quoted the relevant literature to support it.
4.As for the acronyms in line146 and 261 you raised, we have made supplementary explanations after the words or in the footnotes of the table, please confirm.
5.For your question about line313, we have adjusted this part of the statement and made changes in revision mode.
6.As for the conclusion, we have revised it according to your comments, deleted the expression of experimental result data in the conclusion, and enhanced the explanation of the conclusion.
7.In response to your question on LINE 286-293, we hereby explain to you: With regard TO your reference to "IT IS IMPORTANT TO PUT UP A TABLE OF VARIATION OF THESE VALUES. AND ABOVE 20%, WHAT DOES IT MEAN? For this problem, we have given the RSD of each group of experiments according to the way of legend in the "Results and discussion" section, and added the explanation when the RSD value exceeds 20%. With regard TO your reference to "BEYOND THE USE OF THE STANDARD DEVIATION, WAS NO STATISTICAL TREATMENT GIVEN TO THE DATA OBTAINED? SOME SAMPLING ANALYSIS OR OTHER RELEVANT STATISTICAL PARAMETERS WERE USED. JUSTIFY WHETHER THE WAY IN WHICH THE RESEARCH WAS CONDUCTED IT WAS NOT NEEDED TO USE OTHER STATISTICAL PARAMETERS ". We have explained this problem in 282-287 above. In this experiment, The standard recovery and RSD values can be used to illustrate the accuracy and precision of the method.

Reviewer 2 Report

Comments and Suggestions for Authors

The manuscript titled "Optimization and application of the QuEChERS-UHPLC-QTOF-MS method for the determination of Broflanilide residues in Agricultural soils" By Nie et al., aims to establish an efficient and reliable method for detecting broflanilide residues in soil samples.

The following comments need to be addressed:

- Line 35: Please use chemical instead of environmental.

- Lines 41-42: Please start the sentence with "Broflanilide [3-benzamido .......] (table 1) ...."

- Line 43: I suggest mentioning address of each company.

- Line 156: 5-points or 6-points?

- Line 187: Please make sure that patterns are the same colour for each soil. i.e. pars of 0.1 mg/kg are not the same in colour for s1, s2, s3, and s4.

- Line 219: GCB, please mention the full name with abbreviation in the first time.

- Line 227: Please start the sentence as follow: Broflanilide (101.4 mg) ..... 

- Some titles (tables and figures needs to be in the right place. Please revise.

Comments on the Quality of English Language

Please review the manuscript for some minor typing errors.

Author Response

Firstly, thanks a lot for your comments on this paper, so that this paper can be more refined. We carefully discussed your comments and made the following adjustments:

1.According to your comments, we have corrected the sentence problems in line35, line41-42, line156, line219 and line227 under the revision mode, please confirm!

2.As for the company address you mentioned that line43 needs to add, we have confirmed it and added relevant description in the paper under the revised mode.

3.As for the line187 you mentioned, we have confirmed the icon color in the legend, but have not found the problem you mentioned, please confirm it again. If there is a problem, please kindly inform us of the specific information again, and we will correct the problem immediately!

4.Finally, we apologize for the problem that the icon format is wrong when uploading. We have corrected the icon format and position to ensure that the article is in the correct position. If there is still a mistake, you can download the PDF format for viewing or contact us again, and we will modify the upload again.